# Targeting MERTK and AXL in *EGFR* Mutant Non-Small Cell Lung Cancer

**DOI:** 10.3390/cancers13225639

**Published:** 2021-11-11

**Authors:** Dan Yan, H. Shelton Earp, Deborah DeRyckere, Douglas K. Graham

**Affiliations:** 1Aflac Cancer and Blood Disorders Center, Children’s Healthcare of Atlanta, Department of Pediatrics, Emory University, Atlanta, GA 30322, USA; dyan2@emory.edu (D.Y.); deborah.deryckere@emory.edu (D.D.); 2UNC Lineberger Comprehensive Cancer Center, Department of Medicine, Chapel Hill, NC 27599, USA; shelton_earp@med.unc.edu; 3Department of Pharmacology, School of Medicine, University of North Carolina at Chapel Hill, Chapel Hill, NC 27599, USA

**Keywords:** MERTK, AXL, TAM family, receptor tyrosine kinase, targeted therapy, NSCLC, *EGFR* mutation

## Abstract

**Simple Summary:**

Expression of MERTK and/or AXL (members of the TAM family of receptor tyrosine kinases) provides a survival advantage for non-small cell lung cancer (NSCLC) cells and correlates with lymph node metastasis, drug resistance, and disease progression. TAM receptors on host tumor infiltrating cells also play important roles in the immunosuppressive tumor microenvironment. Thus, MERTK and AXL are attractive biologic targets for NSCLC treatment, and clinical trials have recently been launched exploring the efficacy of MERTK/AXL inhibitors in NSCLC. This timely review will address the potential clinical impact of these agents as well as potential side effects to be monitored with the use of these novel drugs.

**Abstract:**

MERTK and AXL are members of the TAM family of receptor tyrosine kinases and are abnormally expressed in 69% and 93% of non-small cell lung cancers (NSCLCs), respectively. Expression of MERTK and/or AXL provides a survival advantage for NSCLC cells and correlates with lymph node metastasis, drug resistance, and disease progression in patients with NSCLC. The TAM receptors on host tumor infiltrating cells also play important roles in the immunosuppressive tumor microenvironment. Thus, MERTK and AXL are attractive biologic targets for NSCLC treatment. Here, we will review physiologic and oncologic roles for MERTK and AXL with an emphasis on the potential to target these kinases in NSCLCs with activating EGFR mutations.

## 1. Introduction

Despite intensive efforts over many years, lung cancer remains difficult to treat and is still the leading cause of cancer-related death in both men and women worldwide. Therapeutic regimens for lung cancer have traditionally been based on tumor histology and morphology. Lung cancers are divided into two broad categories: non-small cell lung cancer (NSCLC) and small cell lung cancer (SCLC). These NSCLCs account for 85% of all lung cancers, and are further subtyped into adenocarcinoma (Ad-NSCLC, ~40%), squamous cell carcinoma (Sq-NSCLC, 25–30%), and large cell carcinoma (LCC, 5–10%) [1,2,3,4]. Although surgery is regarded as the best treatment choice for early-stage and locally advanced NSCLC [2,5], only 20–25% of NSCLC tumors are suitable for curative resection [6]. Over 50% of patients are diagnosed with metastasis [7] and these patients traditionally have dismal survival with a high rate of recurrence after resection [8,9,10,11]. Conventional, highly cytotoxic, platinum-based chemotherapies remain the cornerstone of treatment in the advanced setting, resulting in an increase in median survival by only 1.5 months relative to supportive care [12]. Treatment goals for patients with recurrent and metastatic disease are to extend life and maintain quality of life; obviously, there is a critical need for more efficacious therapies [13]. As the 5-year overall survival rate decreases from 68% in patients with stage IB disease to less than 10% in patients with stage IVA-IVB disease [2,14], early detection, particularly for high risk patients, may also enhance survival.

Following clinical validation of translational inhibitors targeting two important NSCLC oncogenic drivers, epidermal growth factor receptor (EGFR) [15,16,17,18] and anaplastic lymphoma kinase (ALK) [19,20,21], molecular-targeted therapies have been applied to the management of metastatic NSCLC, resulting in remarkably improved prognosis and quality of life relative to patients treated with conventional chemotherapeutics [22,23,24]. Additional mutated oncogenic proteins have been identified in NSCLC, including *HER2*, *BRAF*, *RET*, *MET*, and *ROS1* [20,25]. Even though patients respond to targeted therapies initially, the majority of patients, if not ultimately all patients, relapse within 1 to 2 years when treated with targeted therapies [26,27,28,29,30]. Therefore, understanding the mechanisms of primary and secondary resistance to current targeted therapies is critical to enhance patient outcomes. New therapeutic approaches will be required to further enhance outcomes. Both MERTK and AXL, members of the TAM (TYRO3, AXL, and MERTK) family of receptor tyrosine kinases (RTK), are emerging therapeutic targets in NSCLC. This review will highlight both the survival role for these RTKs in tumor cells and the physiologic, anti-inflammatory functions that are subverted in the tumor microenvironment. Thus, MERTK and AXL targeting may have a dual therapeutic action in NSCLC. The potential to target MERTK and AXL in mt*EGFR*-expressing NSCLC to improve clinical outcomes will also be discussed.

## 2. Physiologic Roles for MERTK and AXL

The TAM kinases are structurally unique from other RTK subfamilies, possessing two immunoglobulin-like (Ig) repeats and two fibronectin type III (FNIII) domains in their extracellular region and a conserved intracellular kinase domain with an unusual signature sequence, KW(I/L)A(I/L)ES [31,32] (Figure 1A). Growth Arrest Specific 6 (GAS6) and Protein S (PROS1) are the two best characterized ligands for TAM receptors, although other TAM ligands have been reported including TUBBY [33], Tubby-like protein 1 (TULP-1) [33], and Galectin-3 (LGALS3) [34]. Structurally, both GAS6 and PROS1 contain a N-terminal glutamic acid-rich (Gla) domain, followed by four epidermal growth factor (EGF)-like repeats, and a C-terminal sex hormone binding globulin (SHBG) homology domain comprised of two globular laminin G-like domains (2-LG) [35] (Figure 1B). The Gla domain confers the ability of these ligands to bind phosphatidylserine (PtdSer) through their N termini. The complex physiologic role of this signaling system is in part defined by this PtdSer sensing which, for example, recognizes the billions of cells that die by apoptosis in the human body daily. However, GAS6 and PROS1 bind differentially to the TAM receptors. MERTK and TYRO3 are activated by both GAS6 and PROS1, while AXL is only activated by GAS6 [36,37]. In the absence of apoptotic cells or PtdSer, the affinity of GAS6 for AXL is more than 6-fold higher than for TYRO3 and 70-fold higher than for MERTK [37,38,39] (Figure 1C). GAS6 binding to both MERTK and TYRO3 is enhanced in the presence of PtdSer, while AXL binding is not stimulated.

Although TAM receptors are expressed in embryonic tissues [32,40,41], triple knockout mice are viable without obvious developmental defects at birth [42], indicating that TAM receptors are not required for embryogenesis. However, knockout of single and/or all 3 TAM receptors is associated with diverse phenotypes in mice, including impaired clearance of apoptotic cells [43], enlarged spleen [44], increased inflammation [45], impairment of tumor cell killing by NK cells [46], hyperproliferation of B and T cells [42], hyperactivation of antigen-presenting cells [42], increased autoantibody production [42,43,47,48], auto-immunity [48], defects in platelet aggregation [49], aborted spermatogenesis and germ cell death [44], neurological abnormalities [44], multiple organ defects, and blindness in adult triple knockout mice [44]. Many of these consequences of TAM receptor loss are related to apoptotic cell clearance and post receptor anti-inflammatory action. This knowledge has spurred research to define the role of TAM RTKs as innate immune checkpoint genes (i.e., guardians against persistent or inappropriate inflammation). Absence of MERTK expression is associated with increased DC activation upon encounter with apoptotic cells, resulting in upregulation of costimulatory molecules and T cell activation [50]. Although some of these phenotypes were noted with knockout of single TAM receptor, they are much more pronounced in triple TAM knockout mice, indicating at least some overlap in TAM kinase functions [43,44,49]. 

## 3. Oncogenic Roles for MERTK and AXL

### 3.1. Roles in NSCLC

MERTK and AXL are frequently aberrantly expressed in NSCLC patient samples, but are absent or expressed at low levels in normal human bronchial epithelial cells [51,52,53,54,55,56,57,58]. High levels of AXL have been described in subsets of both treatment-naïve and relapsed NSCLC [58,59,60]. Increased AXL expression was associated with increased tumor cell invasiveness and tumor grade and predicted poorer survival in patients with NSCLC [56,57,61,62,63,64,65]. Inhibition of MERTK in NSCLC cell lines with a small molecule MERTK tyrosine kinase inhibitor (TKI), MERTK-specific blocking monoclonal antibody, or shRNA induced apoptosis and decreased colony formation in vitro and inhibited tumor growth in vivo [66,67,68,69]. Treatment with an antibody against active AXL or siRNA/shRNA AXL knockdown also provided anti-tumor activity in NSCLC models [56,64,70]. 

High levels of MERTK and/or AXL have also been implicated in drug resistance and radioresistance [54,55,60,69,71,72,73,74,75,76,77,78,79,80,81,82,83,84,85,86]. Increased MERTK or AXL expression in NSCLC correlated with chemotherapy resistance [51,66,70,87,88,89]. Conversely, MERTK or AXL knockdown, treatment with a MERTK monoclonal antibody, or AXL inhibitor R428 or MP-470 promoted apoptosis and increased the sensitivity of NSCLC cells to chemotherapeutic agents [51,66,70,87,88,89]. Upregulated AXL and its interaction with EGFR were associated with resistance to PI3Kα inhibition due to sustained mTOR activation, and addition of AXL inhibitor R428 sensitized tumor cells to PI3Kα [76]. In addition, AXL has been implicated in resistance to anti-IGF-1R therapy [90,91] and resistance to BRAF/MEK inhibitors [92]. Similarly, high levels of AXL protein were associated with resistance to ATR inhibition and treatment with AXL inhibitor R428 sensitized NSCLC cells to ATR inhibitors, VX-970 and AZD6738, resulting in significant DNA damage [59]. Further, AXL expression was associated with JAK1-STAT3 signaling in treatment-naïve tumors and lung cancer patient-derived organoids with high levels of AXL and JAK1 were sensitive to combined treatment with AXL kinase inhibitor TP-0903 and JAK inhibitor ruxolitinib [93]. In addition, MERTK expression was increased when NSCLC cell lines were treated with AXL inhibitor or AXL expression was inhibited using siRNA and dual inhibition of AXL and MERTK reduced cell expansion in vitro and tumor growth in vivo [78]. These data implicate MERTK and/or AXL as potential therapeutic targets in NSCLC. 

### 3.2. Functions in Cancer Cells

MERTK was cloned from a B ALL cell line cDNA library and is ectopically expressed in over 30–50% of childhood acute lymphoblastic leukemia samples and the majority of lymphoid leukemia cell lines, but was not expressed in normal T and B lymphocytes [32,94]. Similarly, AXL was originally cloned from chronic myelogenous leukemia patient samples but is not expressed in granulocytes or lymphocytes [31], suggesting upregulation in the context of hematopoietic malignancy. Both MERTK and AXL are expressed in a wide variety of human cancers, including NSCLC [51], melanoma [95], leukemia [96,97], breast [98], colon [99], liver [100], gastric [101], prostate [102,103], ovarian [104], and brain cancers [105,106], where they promote tumor cell survival and/or proliferation and contribute to oncogenesis. Expression and stimulation of a chimeric MERTK protein in NIH3T3 cells was sufficient to confer anchorage-independent colony formation, a hallmark of oncogenic transformation [107]. Similarly, expression of MERTK conferred IL-3-independence in Ba/F3 cells [108,109]. These data suggest a tumor promoting role for MERTK. Similarly, overexpression of *AXL* induced transformation of NIH3T3 cells and the resulting cells were tumorigenic in nude mice [31,110]. Although MERTK overexpression in normal lung epithelial cells was not sufficient to drive tumorigenesis in vivo, it promoted expansion of normal lung epithelial cells in culture and enhanced clonogenic potential [55]. 

TAM receptors, particularly AXL, have been associated with epithelial–mesenchymal transition (EMT) [87,111,112,113,114]. EMT is an important step in the development of metastatic disease in which cell–cell contacts are lost, leading to tumor cell migration, invasion, and metastasis, and has been associated with therapeutic resistance in NSCLC [87,115,116,117,118,119]. Aberrant expression of AXL promotes phenotypes associated with EMT and metastasis, and inhibition of AXL reduces indicators of EMT/metastasis in various cancers [65,112,120,121,122]. Ectopic overexpression of AXL in NSCLC cell lines caused increased filopodia formation, while silencing of endogenous *AXL* led to loss of spindle-like morphology [123]. NSCLC cells that express high levels of AXL generally expressed abundant vimentin, a transcriptional regulator that contributes to EMT phenotypes [87], and downregulation of AXL decreased expression of vimentin in NSCLC [74,87,112]. Further, ectopic expression of AXL increased migration and invasion in NSCLC cells and AXL inhibition reduced the invasive capacity of NSCLC cell lines [61,64,123]. Enhanced metastasis was accompanied by AXL-dependent MMP-9 activation [124]. Similarly, overexpression of MERTK promoted migration in both normal lung epithelial and NSCLC cell lines [55]. 

Cancer stem cells (CSCs) are a small subpopulation of self-renewing cells within the tumor that influence therapeutic resistance, recurrence, and metastasis [125,126]. Some reports suggest a role for TAM receptors in cancer stemness [125]. MERTK was upregulated in glioblastoma multiforme (GBM) stem-like cells and silencing of MERTK suppressed the self-renewal of patient-derived GBM stem-like cells [127]. Maintenance of the CSC phenotype was mediated by STAT3/KRAS/SRC signaling downstream of MERTK. AXL has also been linked to cancer stemness. Expression of *AXL* positively correlated with expression of several stem cell genes (e.g., *Isl1*, *Cdc2a*, *Bglap1*, *CD44*, and *ALDH1*), resulting in tumorigenicity and chemoresistance [65,122,128]. Interestingly, expression of several other stem cell genes (e.g., *Smad9*, *S100b*, *Mme* and *Col1a1*) negatively correlated with expression of *AXL*. Similarly, knockdown of MERTK, but not AXL or TYRO3, up-regulated stemness-associated genes in dormant prostate cancer cells [129]. Thus, the TAM kinases may have varying roles in cancer stemness, and further study is needed to understand their specific roles in lung CSCs. Prostate CSCs expressed high levels of GAS6, suggesting that MERTK and/or AXL functions in cancer stem cells are related to their kinase activity [130].

### 3.3. Signaling in Cancer Cells

Activation of AXL and/or MERTK leads to signaling cascades that are important for tumor progression (Figure 2). MERTK kinase activity is associated with phosphorylation at three tyrosine residues: Y749, Y753, and Y754 on MERTK [131] and Y779, Y821, and Y866 on AXL [132,133]. Both Y779 and Y821 on AXL and two additional phosphorylation sites for MERTK (Y872 and Y929) are docking sites for GRB2 and the p85 regulatory subunit of PI3K, which activate MEK/ERK and PI3K/AKT signaling pathways, respectively [109,132,133]. The MEK/ERK signaling is associated with cell proliferation [134], while the PI3K/AKT pathway is preferentially involved in tumor cell survival [135]. MERTK-dependent cell migration is mediated by FAK signaling [55,136], while MERTK induced transformation correlates with activation of STAT-dependent transcription [137]. The anti-apoptotic effects of MERTK also correlate with negative regulation of the pro-apoptotic tumor suppressor WW domain-containing oxidoreductase (Wwox) [138]. Also, AXL inhibition is reported to mediate apoptosis by reducing the expression of the anti-apoptotic protein MCL1 [139]. AXL dimerizes with and phosphorylates EGFR to promote activation of the PLCγ-PKC-mTOR signaling cascade and tumor cell survival [76]. Similarly, there is crosstalk between MERTK and EGFR and they are frequently co-expressed on both mt*EGFR*- and wt*EGFR*-expressing NSCLC cell lines [69,80]. In fact, MERTK stabilized the EGFR protein on the cell surface, probably by preventing EGFR internalization and degradation, as EGF-dependent EGFR turnover was reversed by inhibition of lysosomal hydrolase activity [140]. Further, inhibition of MERTK expression using siRNA destabilized expression of EGFR protein.

### 3.4. Immune Regulatory Functions in the Tumor Microenvironment

MERTK is upregulated upon monocyte to macrophage differentiation [141,142,143]. Expression of MERTK and AXL on tumor-infiltrating macrophages polarizes them towards a pro-tumor M2-like phenotype [143,144,145,146]. M2 macrophages promote an immunosuppressive tumor microenvironment by increasing expression of wound-healing cytokines (IL-10, TGFβ, and IL-4) and decreasing pro-inflammatory cytokines (IL-12, TNFα, and IL-6) [45,143,147,148,149,150,151] (Figure 3A). MERTK activation negatively regulates the secretion of pro-inflammatory cytokines, such as TNFα, through suppression of NFκB activation in macrophages [45,152]. LPS challenge led to over-produced TNFα in *Mertk*^kd^ mice, which lack the tyrosine kinase signaling domain, due to hyper-activation of NFκB [45,153]. Inhibition of MERTK by knockout of *Mertk* in mice, neutralization of TAM kinase signaling using a recombinant MERTK-Fc protein as a ligand sink or a GAS6 blocking antibody, and knockout of AXL in macrophages also impaired M2-macrophage anti-inflammatory phenotypes, decreased immunosuppressive IL-10 production, and increased pro-inflammatory IL-12 release [86,143,154,155]. These cytokine alterations lead to expansion of anti-tumor CD8^+^ T lymphocytes and inhibition of tumor growth and metastasis (Figure 3B). Indeed, inhibition of MERTK in the tumor microenvironment in *Mertk*^−/−^ mice was sufficient to decrease tumor growth and metastasis [154]. MERTK-expressing dendritic cells can also regulate T cell activation directly [156]. Blocking MERTK on dendritic cells using anti-MERTK antibody promoted T cell proliferation, while treatment with a MERTK-Fc protein to mimic the effect of MERTK expressed on human dendritic cells suppressed naïve CD4^+^ T cell proliferation [156]. The anti-inflammatory effect of MERTK activation in macrophages and apoptotic cell-treated dendritic cells was mediated by inhibition of NFκB activation [50,157,158] or by induction of toll-like receptor (TLR) suppressor of cytokine signaling 1 (SOCS1) and SOCS3 [158,159,160] (Figure 2). Further, the MERTK ligand PROS1 also promotes resolution of inflammation by macrophages and inhibits macrophage M1 polarization to reduce anti-tumor immune response [161]. More recently, MERTK blockade using anti-MERTK antibody induced a rapid local type I IFN response in tumors [162]. The type I IFNs in turn upregulated the TAM receptors through IFNAR-STAT1 signaling and the upregulated TAM system hijacked the IFNAR-STAT1 cassette to induce the cytokine and TLR suppressors SOCS1 and SOC3 [158,163,164,165,166,167] (Figure 3A). AXL knockout in tumor cells also promoted antigen presentation through increased MHCI expression, leading to an enhanced CD8^+^ T cell response [71]. Furthermore, treatment with the pan-TAM kinase inhibitor sitravatinib reduced tumor burden via activated innate and adaptive immune cells [168]. Additionally, treatment with sitravatinib converted immunosuppressive M2-type macrophages to immunostimulatory M1-type macrophages [168], and this effect was dependent on MERTK expression in bone marrow derived macrophages [158,168]. These findings support roles for MERTK and AXL as tolerogenic receptors that mediate immunosuppression in the tumor microenvironment [143,151,167,169].

Enhanced TAM receptor signaling in response to PtdSer expressed on apoptotic cells resulted in AKT-dependent PD-L1 expression on tumor cells and macrophages, and MERTK inhibition by genetic deletion or treatment with MERTK inhibitor MRX-2843 led to decreased expression of PD-L1 on tumor cells and innate immune cells [170]. In turn, T cell function was indirectly suppressed [38,170,171] (Figure 3B). *AXL* expression was positively correlated with PD-L1 and CXC chemokine receptor 6 (CXCR6) expression in lung cancer, especially in mt*EGFR*-expressing NSCLC [172,173]. Similar to MERTK inhibition, treatment with AXL inhibitor R428 decreased mRNA expression of *PD-L1* and *CXCR6* in mt*EGFR*-expressing NSCLC [172]. In contrast, increased expression of AXL coincided with reduced overall survival in patients treated with PD-1 blockade [172,173]. Accordingly, high levels of AXL expression in lung cancer cells correlated with intrinsic resistance to killing by both natural killer cells and cytotoxic T lymphocytes and this phenotype could be reversed by treatment with the AXL inhibitor R428 [174]. MERTK plays a role in phagocytosis of apoptotic cells in macrophages, but not in dendritic cells [43,142,175]. In contrast, AXL has a greater role in DCs and a lesser role in apoptotic cell phagocytosis by macrophages [176]. Recently, Zhou et al. found that MERTK blockade on tumor-associated macrophages led to accumulation of dying or dead cells in the tumor, resulting in a large increase in extracellular ATP when cells became necrotic [162,177,178]. The increased extracellular ATP in turn opened the ATP-gated P2 × 7R channel and allowed tumor-derived extracellular cGAMP to reach the cytosol of immune cells to activate the adaptor protein stimulator of interferon genes (STING), which in turn triggered the TANK-binding kinase 1-interferon regulatory factor 3 (TBK1-IRF3)-dependent signaling process, leading to the production of type I IFNs [162,179,180,181,182,183]. Cyclic GAMP-AMP synthase (cGAS)-STING signaling in immune cells is a key determinant for therapeutic efficacy of immune checkpoint inhibitors [182,184]. Indeed, blockade of MERTK or AXL using a specific antibody or treatment with sitravatinib or R428 synergized with anti-PD-1 or anti-PD-L1 therapy to enhance anti-tumor immune responses [86,162,168]. 

Immunotherapies, including checkpoint inhibitors, are making an impact as monotherapy and in combination [185]. In a clinical trial in patients with advanced NSCLC without activating *EGFR* or *ALK* mutation and with PD-L1 expression on greater than 50% of tumor cells, pembrolizumab increased response rate (45% vs. 28%), progression-free survival (PFS, 10.3 vs. 6 months) and overall survival (30 vs. 14.2 months) relative to patients treated with chemotherapy, establishing pembrolizumab as the standard of care for these patients [186]. Further work is necessary to explore specific mechanisms of primary and adaptive resistance. 

## 4. Targeting TAM Kinases and EGFR in NSCLC

Although most naïve mt*EGFR*-expressing NSCLC cells are initially sensitive to EGFR TKI treatment [15,17,187,188], acquired resistance eventually happens [189,190,191,192,193,194,195] and AXL and/or GAS6 are frequently upregulated [53,58,196,197]. Other mechanisms that drive secondary resistance to EGFR TKIs are summarized in Table 1. MERTK and AXL are co-expressed with EGFR in NSCLC [58,69,80], and both receptors, but especially AXL, have been studied as causative drivers of secondary EGFR TKI resistance in mt*EGFR*-expressing NSCLCs [53,70,112,196]. Forced expression of an active form of AXL, but not a kinase-impaired AXL, in erlotinib-sensitive tumor cells was sufficient to induce erlotinib resistance [53]. AXL or GAS6 knockdown [53], AXL degradation [75,198], treatment with an AXL blocking antibody [70], or treatment with the small molecule AXL inhibitors XL-880 [53], MP-470 [53], SGI-7079 [112], MGCD265 [199], MGCD516 [199], DS-1205b [200], or R428 [199,201] sensitized NSCLC cells to first generation EGFR TKIs (gefitinib or erlotinib) and delayed onset of resistance relative to EGFR TKI alone [200]. Similarly, MERTK overexpression was sufficient to confer erlotinib resistance in a mt*EGFR*-expressing NSCLC cell line and treatment with a MERTK-selective inhibitor re-sensitized the cells to erlotinib [55]. In paired patient samples collected before and after EGFR TKI treatment, co-expression of AXL and GAS6 was increased in 5 of 35 specimens and 5 additional samples exhibited upregulation of GAS6 alone [53].

Osimertinib is currently the preferred treatment choice for NSCLCs with mt*EGFR* due to its superior efficacy compared to earlier generation EGFR TKIs [22,230,236,237,238,239]. However, osimertinib resistance inevitably occurs, and the mechanisms of resistance to osimertinib are not clear in approximately two-thirds of resistant cases [195,197,240]. A more detailed mechanistic understanding is needed to facilitate development of effective approaches to overcome resistance [241]. Upregulation of MERTK and AXL has been observed in osimertinib-resistant mt*EGFR*-expressing NSCLC cell lines, implicating these kinases as mediators of resistance [54,80]. In addition, both MERTK and PROS1 were upregulated in osimertinib-resistant mt*EGFR* NSCLC cell line derivatives and in mt*EGFR* NSCLC xenograft tumors treated with osimertinib. Moreover, treatment with MERTK kinase inhibitor MRX-2843 [67,242] re-sensitized resistant cells to osimertinib in cell-based assays and provided durable tumor regression in a mt*EGFR* NSCLC xenograft model, even after treatment ended [80]. NSCLCs expressing mt*EGFR* and high levels of AXL were more tolerant to osimertinib treatment than tumors expressing low levels of AXL [54,241]. Treatment with an AXL degrader [198] or an AXL inhibitor DS-1205b [200], ONO-7475 [82], or NPS-1034 [54] reversed osimertinib resistance. Additionally, increased IGF-1R signaling was observed in AXL-low mt*EGFR*-expressing NSCLC tumor cells treated with osimertinib and knockout of IGF-1R sensitized AXL-low NSCLC cells to osimertinib, suggesting that IGF-1R drives cell survival in the presence of osimertinib [241]. Indeed, transient IGF-1R inhibition combined with continuous osimertinib eradicated tumors and provided durable tumor growth inhibition even after cessation of osimertinib [241,243]. 

AXL overexpression and activation also conferred resistance to EGFR antibody cetuximab in NSCLC models both in vitro and in vivo and AXL knockdown decreased EGFR phosphorylation in a cetuximab-resistant NSCLC cell line derivative, but not in parental cells [72]. EGFR knockdown led to a loss of total AXL protein and mRNA expression in both parental and cetuximab resistant cells, indicating additional levels of interplay. Treatment with an anti-AXL monoclonal antibody reduced both AXL and total EGFR, while treatment with AXL inhibitor R428 led to a loss of EGFR phosphorylation in cetuximab-resistant cells, but not in parental cells. As a result, cetuximab-resistant NSCLC cells were sensitive to anti-AXL antibody or R428 treatment. Similarly, MERTK and EGFR are frequently co-expressed in wt*EGFR*-expressing NSCLC cell lines [69]. 

Nearly 90% of *EGFR* mutations found in NSCLC tumors are due to deletions in exon19 or a point mutation in exon21 (L858R) [244]. In patients with NSCLCs that express low levels of *AXL*, median PFS was significantly longer in patients with *EGFR* exon 19 deletion (28.8 months) than in patients with NSCLCs harboring an *EGFR* L858R mutation (9.1 months) and patients with wild-type *EGFR* (11 months) (*p* < 0.0001) [245]. In patients with NSCLCs that express high levels of *AXL*, there was no significant difference in PFS time between the subtypes of mt*EGFR* (*p* > 0.05). Consistent with these findings, high levels of AXL correlated with poor response to initial EGFR TKIs in patients with mt*EGFR* NSCLC [54,65]. A subpopulation of AXL-positive cancer cells were found in erlotinib-naïve tumors [60], providing rationale to treat mt*EGFR*-expressing NSCLCs with AXL and EGFR TKIs combined in order to delay or prevent resistance. 

## 5. MERTK and AXL Inhibitors for Potential Use in NSCLC 

Several RTK inhibitors that target lung cancer abnormalities, such as *EGFR* mutation, *ALK* fusion, *MET* amplification, and *KRAS* mutation, are either in clinical use or currently in clinical trials [237,246,247,248]. Targeting MERTK and AXL may have dual benefits: (1) directly targeting and killing cancer cells and (2) indirectly impacting tumor growth through modulation of the tumor microenvironment [83,249]. Current translatable inhibitors blocking TAM receptors include biologic agents and small molecule inhibitors (Table 2 and Table 3). 

### 5.1. Biological Agents

Antibodies are likely to be the most specific inhibitors of MERTK or AXL with reduced off-target effects (Figure 3B). Treatment with inhibitory anti-AXL monoclonal antibodies YM327.6S2 or 12A11 inhibited tumor growth in a NSCLC xenograft model [64]. The AXL antibody hMAb173 induced apoptosis of renal carcinoma cells and inhibited tumor growth by 78% in vivo [250]. Two additional anti-AXL mAbs (D9 and E8) decreased AXL expression by promoting AXL internalization, resulting in inhibition of proliferation and migration in vitro and reduced tumor growth in murine models [251]. Similarly, the anti-MERTK monoclonal antibody Mer590 reduced surface MERTK protein levels by 87% and increased chemosensitivity and reduced colony formation in NSCLC cell lines [66]. Antibodies are likely to be the most specific inhibitors of MERTK or AXL with reduced off-target effects (Figure 3B). Treatment with inhibitory anti-AXL monoclonal antibodies YM327.6S2 or 12A11 inhibited tumor growth in a NSCLC xenograft model [64]. The AXL antibody hMAb173 induced apoptosis of renal carcinoma cells and inhibited tumor growth by 78% in vivo [250]. Two additional anti-AXL mAbs (D9 and E8) decreased AXL expression by promoting AXL internalization, resulting in inhibition of proliferation and migration in vitro and reduced tumor growth in murine models [251]. Similarly, the anti-MERTK monoclonal antibody Mer590 reduced surface MERTK protein levels by 87% and increased chemosensitivity and reduced colony formation in NSCLC cell lines [66]. 

In addition to anti-TAM specific antibodies, ligand sinks have been developed to block activation of the TAM kinases. GL21.T, a selective RNA-based aptamer, bound to the ectodomain regions of AXL (Kd = 12 nM) and reduced catalytic activity and AXL-dependent signaling, inhibited cellular migration and invasion, and reduced NSCLC tumor volume by 68.2% in vivo [255,275]. Extracellular domains of MERTK and AXL fused to the Fc domain of immunoglobulin G1 can also titrate GAS6 and thereby block GAS6-dependent signaling through the TAM kinases, resulting in decreased tumor metastasis [104] (Figure 3B). Several engineered AXL decoy receptors with 80-fold higher affinity for GAS6 compared to the wild-type AXL receptor significantly reduced tumor burden and metastasis in vivo [254,276,277]. Antibody-drug conjugates, AXL-Fc fusion proteins, and CAR-T therapies directed against the TAM receptors are in clinical trials (Table 2).

### 5.2. Small Molecule Inhibitors

Several small molecules targeting MERTK and/or AXL are in preclinical development. Both AXL and the structurally related kinase MET are frequent mediators of resistance in NSCLC and targeting both may be particularly effective to prevent resistance [212,278,279]. MET and AXL share similar ATP binding sites so small molecule inhibitors targeting MET often inhibit AXL as well [280]. Treatment with NPS-1034, a dual AXL (IC50 = 10.3 nM) and MET (IC_50_ = 48 nM) kinase inhibitor, overcame resistance to EGFR TKIs associated with MET and/or AXL abnormalities [73]. Treatment with LDC1267, a potent and selective TAM kinase inhibitor, enhanced anti-metastatic NK cell activity in vivo [281]. UNC2025 and MRX-2843 are structurally related and orally available MERTK-selective inhibitors suitable for clinical application [67,242]. Both compounds induced apoptosis and decreased colony formation in MERTK-dependent NSCLC cell lines and reduced tumor growth in murine models [68,69]. MRX-2843 also enhanced inhibition of downstream oncogenic signaling in combination with third-generation EGFR TKIs, including osimertinib, resulting in more robust anti-tumor activity both in vitro and in vivo [69,80]. Based on these findings, MRX-2843 is currently in phase I/Ib clinical trials, including an ongoing trial testing MRX-2843 and osimertinib for treatment of advanced mt*EGFR*-expressing NSCLC (Clinicaltrials.gov, NCT04762199). Translational TAM kinase inhibitors also have demonstrated immune-mediated therapeutic activity in preclinical models. Treatment with a pan-TAM TKI RXDX-106 increased tumor-infiltrating leukocytes, M1-polarized intra-tumoral macrophages, and natural killer cell activation and decreased tumor growth [249]. UNC2025 decreased anti-inflammatory M2 macrophages in glioblastoma [282]. MRX-2843 decreased PD-L1 and PD-L2 expression on CD11b^+^ monocytes/macrophages and indirectly decreased PD-1 expression on T cells, leading to increased T cell activation in the tumor microenvironment [170]. 

Because the TAM kinases are closely related, it is likely that many of the inhibitors that were not specifically designed for selectivity are not selective amongst the TAM kinases. This lack of specificity may be an advantage for some applications. The TAM kinases have overlapping functions and they may provide bypass signaling to compensate for selective inhibition of other family members. For instance, MERTK is upregulated in NSCLC cell lines in response to AXL inhibition and promotes resistance to AXL inhibitors [78]. Moreover, inhibition of MERTK sensitized NSCLC cells to AXL inhibition. Thus, in at least some cases agents that target both AXL and MERTK may provide better tumor growth control than more selective compounds.

### 5.3. Potential on-Target Toxicities Associated with MERTK and/or AXL Inhibition

Long-term inhibition of MERTK and/or AXL may cause adverse effects correlated with their important physiologic roles. Retinal pigment epithelial (RPE) cells may fail to phagocytose shed outer segments [283], leading to impaired vision. Additionally, defects in phagocytosis and defects in downregulation of the immune response may cause increased production of self-reactive antibodies and development of autoimmunity [43,48,176]. MERTK and AXL are also expressed on platelets and play roles in clot stabilization [49,284,285]. TAM kinase inhibition may therefore lead to reduced clot stability, although bleeding times are not changed in mice with *Mertk* knock-out or treated with UNC2025 [286]. White and red blood cell counts are decreased in mice treated with high doses of UNC2025 or MRX-2843 [170,287]. These potential toxicities will need to be carefully monitored in ongoing clinical trials.

## 6. Conclusions

MERTK and AXL mediate multiple oncogenic phenotypes in lung cancer, including tumor cell growth, survival, metastasis, invasion, and drug resistance, and are potential targets for lung cancer treatment. Early phase clinical trials of MERTK and AXL inhibitors are underway, and their results will inform future trials. Biomarker-driven clinical trials of selective MERTK and/or AXL inhibitors with real-time monitoring of MERTK and/or AXL activity will better define the true potential of targeting MERTK and/or AXL in the clinic to improve patient outcomes.

## Figures and Tables

**Figure 1 cancers-13-05639-f001:**
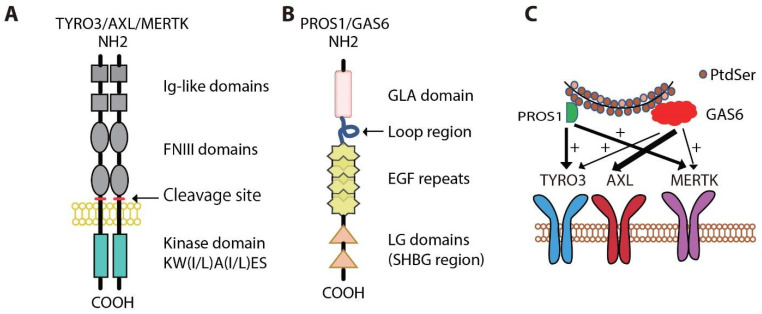
TAM receptors and their ligands. (**A**) TAM receptors TYRO3, AXL, and MERTK share a similar structure of two immunoglobulin (Ig)-like domains, two fibronectin type III (FNIII) domains, and an intracellular kinase domain. (**B**) GAS6 and PROS1 contain a γ-carboxyglutamic acid (Gla) domain, four EGF-like domains, and two lamine G (LG)-like domains. (**C**) Interaction of TAM receptors with their ligands GAS6 and PROS1. The thickness of the arrows indicates the binding strengths of each ligand to the TAM receptors. “+” indicates the enhanced signal in the presence of phosphatidylserine (PtdSer).

**Figure 2 cancers-13-05639-f002:**
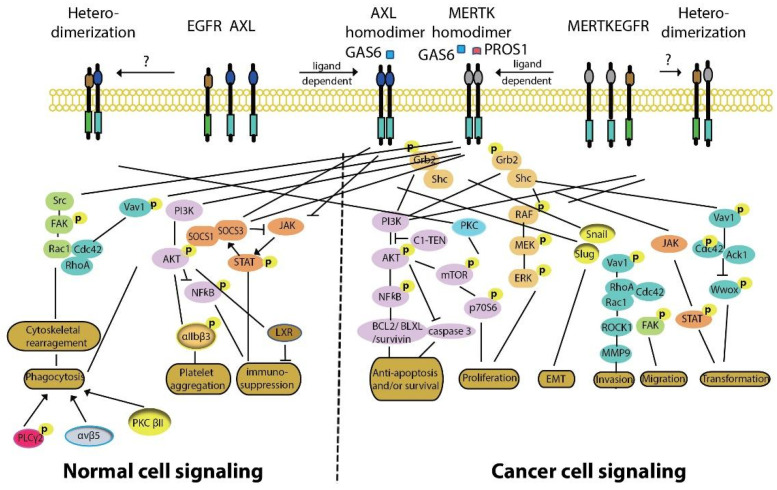
MERTK and AXL signaling in normal and cancer cells. MERTK and AXL play important physiological roles in phagocytosis, platelet aggregation, and immune suppression. Abnormally expressed MERTK and/AXL on NSCLC and other cancer cells are involved in tumorigenesis, including promoting tumor cell survival and proliferation and tumor cell invasion and metastasis. Besides, cross talk between AXL and EGFR, MERTK and EGFR, and AXL and MERTK have also been implicated in drug resistance in the treatment of NSCLC.

**Figure 3 cancers-13-05639-f003:**
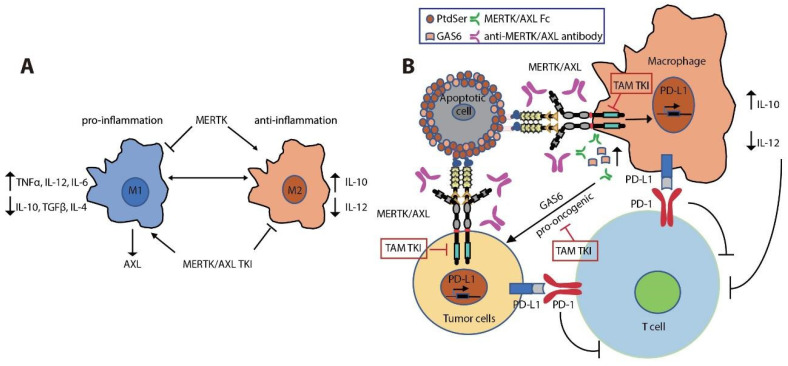
Immune regulatory roles of MERTK and AXL in the TME. (**A**) MERTK signaling favors macrophage to M2 type to generate immunosuppressive microenvironment through releasing anti-inflammatory cytokine IL-10 and decreasing the release of pro-inflammatory cytokines IL-12 and TNFα. While expression of AXL is promoted by pro-inflammatory M1 macrophage, treatment with MERTK/AXL TKI promotes M2 to M1 macrophage development. (**B**) MERTK/AXL signaling favors tumor growth in cancer through two independent mechanisms. Enhanced GAS6 secretion by tumor-associated macrophages promote tumor growth through the activation of oncogenic MERTK/AXL signaling in tumor cells. Activation of MERTK in tumor-associated macrophages and in tumor cells promotes PD-L1 expression, resulting in suppression of T cell activation. Besides, the immunosuppressive cytokine environment limits T cell proliferation and effector functions.

**Table 1 cancers-13-05639-t001:** Mechanisms of resistance to EGFR TKIs independent of TAM kinases.

EGFR TKI	Mechanism of Resistance	Treatment	Reference
Gefitinib/erlotinib	*EGFR* T790M mutation	afatinib, afatinib + rapamycin, CI-387,785, dacomitinib, HKI-272, CO-1686, osimertinib, WK88-1, afatinib + MET inhibitor ARQ 197, PF00299804	[192,195,202,203,204,205,206,207,208]
	*EGFR* T854A mutation	BIBW2992	[209]
	*EGFR* exon 20 mutation		[210]
	EMT	MS-275 + erlotinib	[116]
	*MET* amp/hepatocyte growth factor (HGF) overexpression	PHA-665752 + gefitinib, XL880, crizotinib + afatinib or WZ4002, NPS-1034 + gefitinib or erlotinib	[73,205,211,212,213,214]
	IGF-1R overexpression	AEW541 + gefitinib	[215]
	FGFR1 overexpression/*FGFR3* fusion	PD173074 + afatinib	[216]
	Small cell lung cancer (SCLC) transformation	Standard SCLC treatments	[117,205]
	*HER2* amp/mutation		[205]
	*EGFR* T263P/G719A	afatinib	[217]
	*NRAS* mutation/*BRAF* mutation/*BRAF* fusion	MEK inhibitor AZD6244 + erlotinib or BRAF inhibitor vemurafenib + erlotinib	[218,219]
	Reduced neurofibromin	MEK inhibitor AZD6244 + erlotinib	[220]
	*CCDC6-RET* fusion		[219]
	Glucose metabolism	2-deoxy-D-glucose + afatinib	[221]
WZ4002	*ERK2* amp	MEK inhibitor CI-1040 + WZ4002	[222]
Afatinib	IL-6R/JAK1/STAT3 activation or TGF-β- IL-6 axis activation	Pyridone 6 + afatinib	[223,224,225]
	*FGFR3* fusion		[219]
	*RET* fusion		[219]
	*ALK* fusion		[219]
Osimertinib	*MET* amp/HGF overexpression		[27,226,227,228,229]
	*EGFR* C797S mutation		[27,29,226,227,228,229,230,231]
	*EGFR* C724S mutation	afatinib	[232,233]
	*RET* fusion	RET inhibitor BLU-667 + osimertinib	[27,229,234,235]
	*PCBP2-BRAF* fusion	MEK inhibitor trametinib	[27]
	SCLC transformation		[226,229]
	*FGFR1* mutation/*FGFR1* amp/*FGFR3-TACC3* fusion		[226,229]
	*KRAS* mutation		[229,230]
	*PIK3CA* mutation/*PIK3CA* amp		[226,227,228,229,230]
	*HER2* amp/*HER2* insertion		[227,228,230]
	*BRAF* mutation/*BRAF* fusion		[219,229]
	*EGFR* T790M loss		[229,230]
	*ALK* fusion		[219,234]
	*JAK2* mutation		[230]

**Table 2 cancers-13-05639-t002:** Summary of biological agents for MERTK/AXL.

Compound	Known Targets	Phase	AXL IC_50_	Reference
Monoclonal antibody				
YW327.6S2	AXL-specific	Preclinical	340 ng/mL	[70]
12A11	AXL-specific	Preclinical	~100 ng/mL	[64]
MA b173	AXL-specific	Preclinical	Unk	[250]
D9 and E8	AXL-specific	Preclinical	Unk	[251]
AXL polyclonal antibody	AXL-specific	Preclinical	Unk	[252]
Mer590	MERTK-specific	Preclinical	6.25 ng/mL	[66,136]
Recombinant Protein				
AXL-Fc	AXL, MERTK, TYRO3	preclinical	Unk	[37,253]
MERTK-Fc	AXL, MERTK, TYRO3	Preclinical	Unk	[37]
Decoy receptor				
AXL “decoy receptor”	GAS6	Preclinical	0.5 mg/kg	[254]
Aptamer GL21.T	AXL-specific	Preclinical	13 nM (Kd)	[255]
Antibody-drug conjugate	AXL	I/II		[256]
BA3011/CAB-AXL-ADC			Unk	
Antibody-drug conjugate	AXL	I/II	0.02–2 µg/mL	[92]
HuMax-AXL-ADC			(in vitro)	
Antibody-drug conjugate	AXL	I/II		[257]
CAB-AXL-ADC			Unk	
AXL-Fc	GAS6	I/II/III		[258]
AVB-S6-500			Unk	
CAR-T	AXL	I/II	Unk	[259]
CCT301-38				

**Table 3 cancers-13-05639-t003:** Summary of small molecule MERTK/AXL kinase inhibitors in clinical trials.

Compound	Known Targets	Phase	AXL IC_50_	MERTK IC_50_	NCT Number	References
MRX-2843	MERTK, FLT3	I/Ib	15 nM (in vitro)	1.3 nM (in vitro)	NCT03510104 NCT04762199	[67,69,242]
DS-1205c		I	1.3 nM (in vitro)	63 nM (in vitro)	NCT03255083 NCT03599518	[200]
S49076	AXL, MET, EGFR, ISRC, FGFR1/2/3	I/II	7 nM (in vitro)	2 nM (in vitro)	ISRCTN00759419	[260]
ASLAN002 (BMS-777607)	AXL, MERTK, and MET	I/II	1.1 nM (in vitro)	16 nM (in vitro)	NCT01721148 NCT00605618	[261]
LY2801653	AXL, MET, MST1R	I	2 nM (in vitro)	10 nM (in vitro)	NCT01285037	[262]
INCB081776	AXL, MERTK	I	0.61 nM (in vitro)	3.17 nM (in vitro)	NCT03522142	[83]
Sitravatinib (MGCD516)	AXL, MET, RET, TRK, DDR2, KDR, PDGFRA, Kit	I	1.5 nM (in vitro)	2 nM (in vitro)	NCT02219711	[263]
SU14813	FLT3, VEGFR, PDGFR, Kit	I	84 nM (in vitro)	66 nM (in vitro)	NCT00982267	[264]
RXDX106	AXL, MERTK, TYRO3, MET	I	0.69 nM (in vitro)	1.89 nM (in vitro)	NCT03454243	[249]
Bosutinib (SKI-606/PF-5208763)	AXL, Src, AbI, TGFB, BMP	I/II	0.56 µM (in vitro)	Unk	NCT00195260 NCT00319254	[265]
Amuvatinib (MP470)	AXL, c-KIT, PDGFR, FLT3, RAD51, RET	I/Ib/II	<1 µM (in cells)	Unk	NCT00894894 NCT00881166 NCT01357395	[266]
Gilteritinib (ASP2215)	AXL, FLT3	I/II/III	<1 nM (in vitro)	Unk	NCT02014558 NCT02421939 NCT02752035 NCT02927262 NCT02997202 NCT03182244 NCT02561455 NCT02456883	[253,267]
Glesatinib (MGCD265)	AXL, MET, VEGFR	I/II	Unk	Unk	NCT00697632 NCT00975767	[268]
Ningetinib	VEGFR2, MET, AXL, MERTK, FLT3, RON	I/II	<1 nM (in vitro)	Unk	NCT03758287 NCT04577703	[269]
Merestinib (LY2801653)	MET, RON, FLT3, AXL	I/II	2 nM (in vitro)	10 nM (in vitro)	NCT01285037 NCT03027284 NCT02711553	[262]
BGB324 (R428)	AXL	I/II	14 nM (in vitro) <30 nM (in cells)	224 nM (in vitro)	NCT024 24617 NCT02488408 NCT02872259 NCT02922777 NCT03184558 NCT03184571 NCT03649321 NCT03654833	[270]
Crizotinib (PF-02341066)	ALK, MET, ROS1, AXL	Ib/II	0.3 µM	Unk	NCT02034981 NCT02511184	[24]
TP-0903	AXL	I/II	27 nM (in vitro) 222 nM (in cells)	Unk	NCT02729298 NCT03572634	[113]
ONO-7475	AXL, MERTK	I/II	0.7 nM (in vitro)	1 nM (in vitro)	NCT03176277 NCT03730337	[82]
SGI-7079	AXL	II	58 nM (in vitro)	Unk	NCT00409968	[112,271]
Sunitinib (SU11248)	KIT, FLT3, PDGFR, VEGFR2, AXL	II	9 nM (in vitro)	Unk	NCT01499121 NCT01034878 NCT00864721	[272]
Foretinib (GSK1363089/XL880)	MET, AXL, VEGFR2, RON, Tie-2	II	11 nM (in vitro)	Unk	NCT01068587	[273]
Cabozantinib (XL184)	AXL, MET, VEGFR2, RET, Kit, Flt-1/3/4, Tie2	II/III	7 nM (in vitro) 42 nM (in cells)	Unk	NCT01639508 NCT01708954 NCT01866410	[274]

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
