# Peer review of "Targeting MERTK and AXL in EGFR Mutant Non-Small Cell Lung Cancer"

_cancers, 2021, doi:10.3390/cancers13225639_

Round 1

Reviewer 1 Report

The review manuscript by Yan D. and colleagues focused on an interesting topic in a very active area of research to identify novel therapeutic targets to overcome mechanisms of resistance in EGFR-mutant NSCLC. In the first part, the authors detailed the physiological and oncogenic roles of MERTK and AXL, while in the second part of the manuscript the focus is on the development of new agents targeting MERTK and AXL as emerging mediators of resistance to EGFR TKIs.  

It is basically a well-written review, with a logic and clear structure and filled with useful information. Only a few concerns were raised in order to improve the manuscript.

  • Lines 225-232: could the authors explained in more detail the role and interconnection between immunotherapy, targeting PD-1/PD-L1 axis, and MERKT/AXL expression?
  • According to the title the review will focus on MERTK and AXL in EGFR-mutant NSCLC: could section 4 be expanded? e.g. it would be helpful if the authors could discuss main mechanisms of resistance to EFGR TKIs (other than MERTK and AXL) and/or summarize them in a table/figure.
  • Have any interconnections between different EGFR-mutation subtypes and MERKT/AXL expression been reported?
  • It would be more readable if authors could summarize, when available, preliminary clinical results of the phase I/II trials mentioned in the manuscript

Author Response

The review manuscript by Yan D. and colleagues focused on an interesting topic in a very active area of research to identify novel therapeutic targets to overcome mechanisms of resistance in EGFR-mutant NSCLC. In the first part, the authors detailed the physiological and oncogenic roles of MERTK and AXL, while in the second part of the manuscript the focus is on the development of new agents targeting MERTK and AXL as emerging mediators of resistance to EGFR TKIs.  

It is basically a well-written review, with a logic and clear structure and filled with useful information. Only a few concerns were raised in order to improve the manuscript.

  • Lines 225-232: could the authors explained in more detail the role and interconnection between immunotherapy, targeting PD-1/PD-L1 axis, and MERKT/AXL expression?

Thank you for this suggestion. These are expanded as shown in section 3.4.

According to the title the review will focus on MERTK and AXL in EGFR-mutant NSCLC: could section 4 be expanded? e.g. it would be helpful if the authors could discuss main mechanisms of resistance to EFGR TKIs (other than MERTK and AXL) and/or summarize them in a table/figure.

As suggested, the mechanisms related with EGFR TKI resistance and reported strategies to inverse the indicated resistance have been added to Table 1.

  • Have any interconnections between different EGFR-mutation subtypes and MERKT/AXL expression been reported?

Cai et al. found that “With low AXL expression, the median PFS time in NSCLC patients with EGFR exon 19 deletion (28.8 months) was significantly longer than in NSCLC patients harboring EGFR L858R mutation (9.1 months) and in patients with wild-type EGFR (11 months) administrated with conventional chemotherapy (p<0.0001) [1]. There was no significant difference in PFS time among the subtypes of mtEGFR-expressing NSCLC patients with high AXL expression administrated with conventional chemotherapy (p>0.05) [1].” The updated information is included in section 4.

  • It would be more readable if authors could summarize, when available, preliminary clinical results of the phase I/II trials mentioned in the manuscript.

The preliminary results for the clinical trials with MRX-2843 are being prepared for presentation in an upcoming national meeting and a publication will follow.

Please see the tracked changes as attachment.

Reviewer 2 Report

The manuscript "Targeting MERTK and AXL in EGFR mutant non-small cell lung cancer" was aimed to review the basic research and clinical studies of members of the TAM family of receptor tyrosine kinases that are abnormally highly expressed in non-small cell lung cancers (NSCLCs).

Analyzing the data of published studies, the authors describes structure-biochemical and functional features of TAM receptors and discuss oncogenic roles of MERTK and AXL in the regulation of cell survival, EMT and lymph node metastasis, drug resistance, and the immunosuppressive tumor microenvironment in different cancers with emphasis on NSCLCs.

The manuscript gives an overview of the latest findings of the field.  The review also provides an analysis of preclinical and ongoing clinical studies of the effects of MERTK and AXL inhibitors, recombinant proteins, antibodies, decoy receptors, antibody-drug conjugates, and small molecules in lung cancers. This review demonstrates that MERTK and AXL can be attractive and potent biologic targets for NSCLC treatment. Still, long-term inhibition of MERTK and/or AXL may cause adverse effects correlated with their important physiologic roles.

In my opinion, this review article is well-writing and may be interesting and helpful for a wide audience of researchers and clinicians, and university students. The information provided in this manuscript may be useful for further basic and pharmacological research. This manuscript is well organized and comprehensively described. The references were used properly.

I have the following minor comments and suggestions to the manuscript:

  1. Figure 2 needs clarification to improve understanding of MERTK and AXL involvement in different signaling pathways in normal and cancer cells. The legend of Figure 2 does not sufficiently explain the interactions of the depicted signaling pathways in normal and cancer cells.

In the presented scheme, various signaling pathways interact at different levels, but it is unclear from which receptors and how these signaling pathways are initiated.

I didn't understand why NF-kB factors are placed downstream in PI3K / Akt signaling, and what are the differences between their involvement of these signaling pathways in normal and cancer cells?

Author Response

The manuscript "Targeting MERTK and AXL in EGFR mutant non-small cell lung cancer" was aimed to review the basic research and clinical studies of members of the TAM family of receptor tyrosine kinases that are abnormally highly expressed in non-small cell lung cancers (NSCLCs).

Analyzing the data of published studies, the authors describes structure-biochemical and functional features of TAM receptors and discuss oncogenic roles of MERTK and AXL in the regulation of cell survival, EMT and lymph node metastasis, drug resistance, and the immunosuppressive tumor microenvironment in different cancers with emphasis on NSCLCs.

The manuscript gives an overview of the latest findings of the field.  The review also provides an analysis of preclinical and ongoing clinical studies of the effects of MERTK and AXL inhibitors, recombinant proteins, antibodies, decoy receptors, antibody-drug conjugates, and small molecules in lung cancers. This review demonstrates that MERTK and AXL can be attractive and potent biologic targets for NSCLC treatment. Still, long-term inhibition of MERTK and/or AXL may cause adverse effects correlated with their important physiologic roles.

In my opinion, this review article is well-writing and may be interesting and helpful for a wide audience of researchers and clinicians, and university students. The information provided in this manuscript may be useful for further basic and pharmacological research. This manuscript is well organized and comprehensively described. The references were used properly.

I have the following minor comments and suggestions to the manuscript:

 Figure 2 needs clarification to improve understanding of MERTK and AXL involvement in different signaling pathways in normal and cancer cells. The legend of Figure 2 does not sufficiently explain the interactions of the depicted signaling pathways in normal and cancer cells.

After ligand-triggered MERTK/AXL dimerization, PI3K-AKT signaling pathway is activated in normal cell and cancer cells, dependent on where MERTK/AXL is expressed. This figure is intended to emphasize the separate signaling pathways in normal vs. cancer cells. The PI3K-AKT signaling pathway in normal cells has been altered to provide further clarity.

In the presented scheme, various signaling pathways interact at different levels, but it is unclear from which receptors and how these signaling pathways are initiated.

Thank you for this comment. The MERTK role in driving macrophage phagocytosis is updated in Fig2 as well.

I didn't understand why NF-kB factors are placed downstream in PI3K / Akt signaling, and what are the differences between their involvement of these signaling pathways in normal and cancer cells?

After ligand stimulation, activated MERTK further activate PI3K-AKT signaling both in normal especially macrophages and tumor cells. In tumor cells, after AKT activation, the phosphorylated AKT further phosphorylate IκB, resulting in polyubiquitination and degradation by the 26S proteasome [1-4]. In turn, the p65/RelA-p50 dimer translocates into the nucleus to bind to specific NFκB binding sites in the enhancer regions of target genes, such as the anti-apoptotic genes, survivin, BCL2, and BCL-XL, to regulate their transcription [5-14]. In macrophages, MERTK activation negatively regulates the secretion of pro-inflammatory cytokines, such as TNFα, through suppression of NFκB activation in macrophages [15, 16]. LPS challenge led to over-produced TNFα in Mertk kd mice, lack the signaling tyrosine kinase domain, due to the hyper-activated NFκB [15, 17]. It was further found that the inhibition of NFκB activation after MERTK activation in macrophages was through the activated PI3K-AKT pathway, suggesting a differential role of PI3K-AKT pathway in normal cell especially in macrophages versus in tumor cells as discussed above [18, 19]. Inhibition of the PI3K-AKT pathway enhanced LPS-induced NFκB nuclear translocation, in turn TNFα was produced [18] (Fig2). Modifications in text have been made in section 3.3 and 3.4.

  1. Bai D, Ueno L, Vogt PK: Akt-mediated regulation of NFkappaB and the essentialness of NFkappaB for the oncogenicity of PI3K and Akt. Int J Cancer 2009, 125:2863-2870.
  2. Gustin JA, Ozes ON, Akca H, Pincheira R, Mayo LD, Li Q, Guzman JR, Korgaonkar CK, Donner DB: Cell type-specific expression of the IkappaB kinases determines the significance of phosphatidylinositol 3-kinase/Akt signaling to NF-kappa B activation. J Biol Chem 2004, 279:1615-1620.
  3. Ozes ON, Mayo LD, Gustin JA, Pfeffer SR, Pfeffer LM, Donner DB: NF-kappaB activation by tumour necrosis factor requires the Akt serine-threonine kinase. Nature 1999, 401:82-85.
  4. Karin M, Ben-Neriah Y: Phosphorylation meets ubiquitination: the control of NF-[kappa]B activity. Annu Rev Immunol 2000, 18:621-663.
  5. Karin M: Nuclear factor-kappaB in cancer development and progression. Nature 2006, 441:431-436.
  6. Sizemore N, Leung S, Stark GR: Activation of phosphatidylinositol 3-kinase in response to interleukin-1 leads to phosphorylation and activation of the NF-kappaB p65/RelA subunit. Mol Cell Biol 1999, 19:4798-4805.
  7. Romashkova JA, Makarov SS: NF-kappaB is a target of AKT in anti-apoptotic PDGF signalling. Nature 1999, 401:86-90.
  8. Liu ZG, Hsu H, Goeddel DV, Karin M: Dissection of TNF receptor 1 effector functions: JNK activation is not linked to apoptosis while NF-kappaB activation prevents cell death. Cell 1996, 87:565-576.
  9. Van Antwerp DJ, Martin SJ, Kafri T, Green DR, Verma IM: Suppression of TNF-alpha-induced apoptosis by NF-kappaB. Science 1996, 274:787-789.
  10. Wang CY, Mayo MW, Baldwin AS, Jr.: TNF- and cancer therapy-induced apoptosis: potentiation by inhibition of NF-kappaB. Science 1996, 274:784-787.
  11. Karin M, Lin A: NF-kappaB at the crossroads of life and death. Nat Immunol 2002, 3:221-227.
  12. Danial NN, Korsmeyer SJ: Cell death: critical control points. Cell 2004, 116:205-219.
  13. Zong WX, Edelstein LC, Chen C, Bash J, Gelinas C: The prosurvival Bcl-2 homolog Bfl-1/A1 is a direct transcriptional target of NF-kappaB that blocks TNFalpha-induced apoptosis. Genes Dev 1999, 13:382-387.
  14. Sethi G, Ahn KS, Aggarwal BB: Targeting nuclear factor-kappa B activation pathway by thymoquinone: role in suppression of antiapoptotic gene products and enhancement of apoptosis. Mol Cancer Res 2008, 6:1059-1070.
  15. Camenisch TD, Koller BH, Earp HS, Matsushima GK: A novel receptor tyrosine kinase, Mer, inhibits TNF-alpha production and lipopolysaccharide-induced endotoxic shock. J Immunol 1999, 162:3498-3503.
  16. Ito K: Impact of post-translational modifications of proteins on the inflammatory process. Biochem Soc Trans 2007, 35:281-283.
  17. Covert MW, Leung TH, Gaston JE, Baltimore D: Achieving stability of lipopolysaccharide-induced NF-kappaB activation. Science 2005, 309:1854-1857.
  18. Guha M, Mackman N: The phosphatidylinositol 3-kinase-Akt pathway limits lipopolysaccharide activation of signaling pathways and expression of inflammatory mediators in human monocytic cells. J Biol Chem 2002, 277:32124-32132.
  19. Verzella D, Pescatore A, Capece D, Vecchiotti D, Ursini MV, Franzoso G, Alesse E, Zazzeroni F: Life, death, and autophagy in cancer: NF-kappaB turns up everywhere. Cell Death Dis 2020, 11:210.